# Comparison of the Gut Microbiota of Jeju and Thoroughbred Horses in Korea

**DOI:** 10.3390/vetsci8050081

**Published:** 2021-05-11

**Authors:** Taemook Park, Jungho Yoon, Ahram Kim, Tatsuya Unno, Youngmin Yun

**Affiliations:** 1Equine Clinic, Jeju Stud Farm, Korea Racing Authority, Jeju 63346, Korea; taemook7@gmail.com (T.P.); junghoy11@gmail.com (J.Y.); aidia0207@naver.com (A.K.); 2College of Veterinary Medicine and Veterinary Medical Research Institute, Jeju National University, Jeju 63243, Korea; 3Subtropical/Tropical Organism Gene Bank, Jeju National University, Jeju 63243, Korea; 4Faculty of Biotechnology, School of Life Sciences, SARI, Jeju 63243, Korea

**Keywords:** microbial ecology, microbiota, NGS, Jeju horse, Thoroughbred horse

## Abstract

(1) Background: The large intestine of horses is an anaerobic fermentative chamber filled with fibrolytic bacteria that play essential roles in digesting and absorbing nutrients for energy production. Although Jeju horses are a prominent local breed in Korea, few studies have investigated the gut microbiota of Jeju horses; (2) Methods: This study performed sequencing of V3 and V4 hypervariable regions of the partial 16S rRNA genes obtained from horse fecal samples and compared the gut microbiota between Jeju and Thoroughbred horses. Thirty and 24 fecal samples were obtained from Jeju and Thoroughbred horses, respectively; (3) Results: The gut microbiota belonged to 23 phyla and 159 families. Firmicutes and Bacteroidetes were the most abundant and predominant phyla, followed by Verrucomicrobia, Euryachaeota, and Spirochaete. The ratio of Firmicutes to Bacteroidetes (F/B), which is known as a relevant marker of gut dysbiosis, was 1.84 for Jeju horses, whereas it was 1.76 for Thoroughbred horses. Moreover, at the genus level, 21 genera were significantly different between the Jeju and Thoroughbred horses (*p* < 0.05); (4) Conclusions: The Thoroughbred horse’s gut microbiotas had significantly higher diversity than the Jeju horses (*p* < 0.05). In addition, beneficial commensal bacteria that produce short-chain fatty acids thus providing a significant source of energy are also more abundant in Thoroughbred horses. These results provide novel information on the horse gut microbiota and insights for further studies related to the horse gut microbiota.

## 1. Introduction

Horses have evolved for more than 50 million years to graze large amounts of grass constantly by absorbing nutrients from the fiber only, providing low-energy to monogastric non-ruminant animals. Horses obtain energy effectively through fermentation by the gut microbiota in the hindgut. The hindgut serves as a fermentation chamber, where anatomical and physiological properties are favorable for the gut microbiota to degrade and ferment plant-derived polysaccharides [1]. Therefore, horses can digest and metabolize a high-fiber plant diet using the enzymes produced by the gut microbiota [2].

A wide variety of microbial ecosystems exist in the gastrointestinal tract of animals. One gram of the content of a horse colon contains at least one billion microorganisms, including at least seven phyla [3,4] or more than 108 genera of bacteria [5,6], as well as protozoa, archaea, and anaerobic fungi. The intestinal microbiota perform several essential protective, structural, and metabolic functions for host health, such as digestion of complex host-indigestible polysaccharides, pathogen displacement, and synthesis of vitamins [7,8]. The intestinal microorganisms manufacture short-chain fatty acids (SCFAs) to produce nutrients in the mucus and mucosal membranes and promote the regeneration of intestinal epithelial cells [9,10]. The composition of the horse gut microbiota is affected by various factors, including race, gender, heredity, diet, lifestyle, geographical location, stress, obesity, and delivery methods [9,11,12,13].

The study of gut microbiota has generally been conducted using culture-dependent methods [14,15]. These methods are limited in investigating the horse gut microbiota because approximately 70% of the microbiota cannot be cultured in the laboratory [16]. The development of next-generation sequencing (NGS) and bioinformatic tools has overcome the limitations of traditional culture-dependent methods [4,17]. To date, there are only limited numbers of studies on the horse microbiome using NGS compared to those for ruminants. Although several studies were conducted to investigate the relationship between gut microbiota and digestion [1], disease [2,3,18], and exercise [19,20], the roles of the horse gut microbiota are not entirely understood. However, as widely acknowledged, the microbiota play a vital role in maintaining the host’s health, including horses [18].

The Jeju horse is the only indigenous Korean horse breed since the Stone Age. They are inherently rare. Thus, they are conserved well and designated as Korean Natural Monument 347. Recently, Jeju horses have been used as racing horses; thus, the economic value of Jeju horses has increased close to that of Thoroughbred horses. They have adapted to the harsh environment of Jeju Island, such as cold weather and insufficient food for a long time.

On the other hand, Thoroughbred horses are sprint racehorses that have adjusted to the environment in the Jeju Island over a short period [21]. They are provided with not only a high-qualified feed but also systematically controlled specific management. Jeju and Thoroughbred horses are raised in different environments (i.e., feeds and exercise). Some differences in the composition of gut microbiota affected by various external factors are expected. The present study provides information on the fundamental differences in gut microbiota between Jeju and Thoroughbred horses based on the 16S rRNA gene sequence data.

## 2. Materials and Methods

### 2.1. Horse Descriptions and Fecal Sampling

The Institutional Animal Care and Use Committee of Korea Racing Authority approved the animal protocols for this study (KRA IACUC-2009-AEC-2007). Between March and June in 2020, fresh fecal samples were collected from healthy Jeju horses (JJ, *n* = 30) with a mean age of 6.8 ± 3.3 years (10 male, 16 female, and four gelded) and healthy Thoroughbred horses (TB, *n* = 24) with a mean age of 6.2 ± 3.1 years (eight male, 14 female, and two gelded). All horses were raised on 12 different farms on the Jeju Island, Korea. All horses had not experienced any changes in their diet and housing conditions in the recent three months and were carefully selected for minimizing the variations in age, weight, body condition scoring (BCS), soundness, vaccination, deworming, and medication. Jeju horses grazed on paddocks for 24 h, while Thoroughbred horses lived in a stable at night. The Thoroughbreds were provided with concentrated and roughage feed at 2.5 to 3% per body weight every day.

In contrast, the Jeju horses were provided with concentrated feed at approximately 1% per body weight and roughage feed at ad libitum every day. Fecal sampling was performed from the rectum using rectal gloves with sterile lubrication (Kruuse, Denmark) to reduce environmental contamination, as described previously [22]. Rectal swabs were taken from these horses, placed in plastic sealable bags, and stored at −80 °C until DNA extraction.

### 2.2. Microbial Community Analysis

According to the manufacturers’ instructions, the fecal DNA was extracted from swabs using the PowerFecal DNA extraction kit (Qiagen, Hilden, Germany). The V3 and V4 regions of the 16S rRNA gene were amplified by a polymerase chain reaction (PCR) using the 341F and 806R primer sets. Two-step PCR was performed to construct the MiSeq library. Sequencing was done at Macrogen Inc. (Seoul, Korea) according to the manufactures’ instructions. The sequence data were processed using MOTHUR [23] according to the standard operational protocol (protocol as previously described online https://mothur.org/wiki/miseq_sop/ (accessed on 14 April 2021)) with a slight modification of adding singleton removal after the pre.cluster subroutine. Silva.nr_v132 [24] was used for alignment, and RDP [25] version 11.5 was used for taxonomic classification. The operational taxonomic units (OTUs) were assigned using the opti.clust algorithm [26] with a sequence distance of 0.03. MOTHUR was also used to calculate the ecological indices (Chao Ι and Shannon) for the species richness and evenness. Non-metric multidimensional scaling (NMDS) was performed and plotted with ellipses at the 95% confidence level using the vegan R package. PICRUSt2 [27] was used to predict the metabolic activities based on the 16S rRNA gene sequences.

### 2.3. Statistics

An analysis of the molecular variances (AMOVA) was applied to investigate the significant difference between the groups on NMDS plots. Differential abundance analysis was performed using the Linear discriminant analysis Effect Size (LEfSe) [28] and ALDEx2 package in R for OTUs and predicted metabolic activities, respectively. A Wilcoxon rank-sum test was applied to compare the ecological indices. Differences were considered significant at *p* < 0.05.

## 3. Results

### 3.1. α-Diversity Analysis

Thirty healthy Jeju horses were compared with 24 healthy Thoroughbred horses through 16S rRNA sequencing. Figure 1 presents the ecological indices. All samples showed a Good’s coverage greater than 98%, suggesting that sequence depth was sufficient to cover most of the species in the samples (Figure 1A). The results in Figure 1 show that the number of species found in horse gut microbiota ranged from approximately 500 species to 1250 species (Figure 1B), while the evenness ranged from 3.5 to 5.5 (Figure 1C). The results suggest that both the species richness and evenness of the gut microbiota in Jeju horses were significantly lower than those in the Thoroughbred horses (*p* < 0.05).

### 3.2. β-Diversity Analysis

Based on non-multidimensional scaling (NMDS) analysis (Figure 2), the β-diversity showed that the gut microbiota of Jeju horses were significantly different from Thoroughbred horses (*p* < 0.05). Thus, the microbiota structure of each group was substantially different from the other.

### 3.3. Comparison of Microbiota at the Phylum and Family Levels

A comparison of the fecal microbial communities was at the bacterial phylum and family levels (Figure 3). At the phylum level (Figure 3A), the bacterial community compositions were similar in the Jeju and Thoroughbred horses. It is consistent with a previous equine microbiota study showing that Firmicutes and Bacteroidetes are predominant phyla [3,4,29,30]. In this study, Firmicutes (39.2 ± 7.9% and 40.9 ± 6.6%) and Bacteroidetes (21.2 ± 7.6% and 23.2 ± 5.3%) were the abundant and predominant phyla in Jeju and Thoroughbred horses, respectively, followed by Verrucomicrobia, Euryachaeota, and Spirochaete. Firmicutes are critical players in cellulose degradation, and they represent the most common phylum in all mammalian species [31]. Bacteroidetes are reduced in human obesity [9]. These five phyla accounted for at least 86% of the gut microbiota found in 26 phyla in this study.

At the family level (Figure 3B), there was no significant difference between the gut microbiota of the Jeju and Thoroughbred horses. Lachnospiraceae and Bacteroidetes_unclassified were the predominant families, followed by Methanobacteriaceae, Streptococcaceae, Ruminococcaceae, Prevotellaceae, Subdivision 5_unclassified, and Spirochaetacea. These eight families accounted for approximately 60% of the total gut microbiota composed of 170 families in this study.

### 3.4. Comparison of Microbiota at the Genus Level

The relative abundance of each genus in both groups was compared to identify the populations responsible for this difference using the Linear discriminant analysis Effect Size (LEfSe) (Figure 4). The results indicated significant differences for 21 genera (*p* < 0.05). At the genus level, the relative abundance of the 17 genera was significantly lower in Jeju horses than in Thoroughbred horses. These genera included in a genus *Fibrobacter*, which is vital in rumen fiber digestion [15], while the Jeju horses appeared to have more pathogenic bacteria genus (i.e., Synergistaceae_unclassified) than Thoroughbred horses.

Further, the relative abundance of some of the predominant commensal intestinal bacteria, such as *Fibrobacter*, Lachnospiraceae_unclassified, Clostridiales_unclassified, and *Oscillibacter,* was significantly higher in the Thoroughbred horses. At the same time, opportunistic bacterial pathogens, including Spirochaetaceae_unclassified and *Sphaerochaeta*, were also more abundant. On the other hand, the genera significantly less abundant in the Jeju horses included butyrate-producing bacteria (i.e., Clostridium_XIVa and *Oscillibacter*). At the same time, significantly higher abundance was observed for some of the opportunistic pathogens, such as Synergistaceae_unclassified, Eubacteriaceae_unclassified, and *Akkermansia*, is known to increase with gut inflammation (*p* < 0.05). These results suggest that Thoroughbred horses are more suitable for sprint racehorses because of the high diversity and abundance of beneficial bacteria producing SCFAs, including acetate, butyrate, and propionate. In contrast, Jeju horses showed a tendency to have more pathogens than Thoroughbred horses.

## 4. Discussion

While the hindgut itself serves as a fermentation chamber, the gut microbiota play vital roles in digesting and absorbing nutrients for the host energy production and synthesizing Vitamin B and K. Moreover, the gut microbiota regulate gene expression in the intestinal epithelial tissues of the host and maintain several diseases, including gastrointestinal diseases [32].

The composition of gut microbiota is affected by various factors, such as surrounding environments, diet, gestation and hospitalization period, antibiotics use, and the mode of delivery [18]. In cold climates, horses are usually provided with low-quality feed, which may adjust their gut microbiota to ferment fibers [33]. This study showed that the bacterial community compositions among Jeju horses were significantly different from those of Thoroughbred horses. Moreover, Jeju horses had lower species diversity than Thoroughbred horses.

Significant differences in bacterial compositions were not observed between Jeju and Thoroughbred horses at the phylum and family levels. Although Jeju and Thoroughbred horses are provided different feed and are raised differently, the proportion of Firmicutes (JJ, 21.2 ± 7.59%; TB, 23.2 ± 5.34%) and Bacteroidetes (JJ, 21.2 ± 7.59%; TB, 23.2 ± 5.34%) were similar among the two groups, which is also consistent with previous studies [3,6]. The ratio of Firmicutes to Bacteroidetes (F/B), which is a relevant marker of gut dysbiosis for humans and mice, was 1.84 and 1.76 for Jeju and Thoroughbred horses, respectively. These ratios were similar for Mongolian horses (F/B = 1.6) but different in Irish Thoroughbred horses (F/B > 2) [17,30], suggesting that Jeju horses may be more similar to Mongolian horses. Hence, the predominant Firmicutes phylum is not surprising because most of them are plant cellulose degraders [15,34].

At the family level, Lachnospiraceae and Ruminococcaceae are the core microbiota in the human intestine and are referred to as healthy microbiota [35]. Lachnospiraceae are gram-positive anaerobes that assist in the digestion of indigestible polysaccharides and the production of short-chain fatty acids (SCFAs) butyrate and propionate. SCFAs help prevent colon cancer and improve intestinal health [36,37]. Ruminococcaceae also breaks down fiber effectively in the intestine of herbivores [29] and produces butyrate in the gut [38]. Prevotellaceae is abundant in ruminant animals (i.e., cattle) and degrades proteins and carbohydrates, which iss abundant in horses living in pastures [39]. Spirochaetaceae and Streptococcaceae are known opportunistic pathogens. They are indicative of intestinal dysbiosis.

At the genus level, several genera with differential abundances were observed. Thoroughbred horses had a higher abundance of various beneficial commensal bacteria than Jeju horses, such as Clostridium_XIVb and Lachnospiraceae_unclassified, which produce butyrate as the primary energy source for colonocytes [40,41], *Fibrobacteres* which digest plant fibers in the herbivore intestine [15], and *Ruminococcus2* from the Ruminococcaceae family, a cellulose-degrading bacterium in human intestine [42].

The previous study reported that the abundance of Clostridium_XlVb and *Bacteroides* increased, but *Ruminococcaceae* decreased in the horses fed a high concentrated diet [43]. However, in this study, even though Thoroughbred horses were fed more concentrated feed than Jeju horses, the abundance of Ruminococcaceae increased significantly in Thoroughbred horses (*p* < 0.05). Kim et al. suggested that the increased abundance of Clostridium_XIVa could adversely affect the abundance of other polysaccharide-utilizing bacteria, such as *Lactobacillus* [44], due to competition over the substrate availability.

In this study, an increase in Lactobacillales_unclassified was observed among Jeju horses. These bacteria may represent some probiotic strains. On the other hand, several opportunistic pathogens, such as Eubacteriaceae_unclassified and Synergistaceae_unclassified, were also higher in Jeju horses. The genus *Akkermansia* from Verrucomicrobia increase is speculated to result from inflammation because of its potential to induce the immune regulatory response [45]. A decrease in bacterial richness and diversity in Jeju horses compared to Thoroughbred horses was one of the main findings of this study. Thoroughbred horses also showed more significant variability between individuals in species richness and diversity than Jeju horses. These differences in gut microbiota could be owing to types of breeds, feed, and management program [30,46]. Previous studies have reported that parasite, exercise, age, and stress significantly shifts horse gut microbiota [19,47,48,49]. Exercise and aging increased the abundance of Firmicutes while stress decreases the abundance of Firmicutes [19,20,49]. Various factors could affect the abundance of several genera. Although we minimized these factors by selecting horses similar in physiology, it is recommended that future studies include a bigger sample size and various physiological and environmental parameters.

In this study, we collected horse feces. Although feces were rectally collected using rectal gloves [22] and stored immediately on ice, storage conditions may have affected the microbial community, as previously reported [50]. Moreover, multiple sampling of the same horses could minimize inter-individual deviation.

## 5. Conclusions

This study compared the compositions of the gut microbiota in Jeju and Thoroughbred horses living on the Jeju Island in Korea. The results showed that the bacterial community composition in Jeju horses was substantially different from that of Thoroughbred horses. Thoroughbred horses had more species and a more diverse bacterial population than Jeju horses do. Several genera were found to be differentially abundant at the genus level between the two groups, where Thoroughbred horses appeared to have more beneficial bacteria than Jeju horses.

Thoroughbred horses receive a relatively large amount of concentrated feed under considerable stress because people manage them. In contrast, Jeju horses freely graze in a vast paddock and live their lives in their natural habit. Nevertheless, Thoroughbred horses appeared to have better gut microbiota than Jeju horses. This might be because of the high-quality diet supplied to Thoroughbred horses. The diet contained well-adjusted and balanced nutrients with essential vitamins, which were consistently provided. Overall, this study presented some insights into the association of gut microbiota with the breed, feed, and management conditions.

## Figures and Tables

**Figure 1 vetsci-08-00081-f001:**
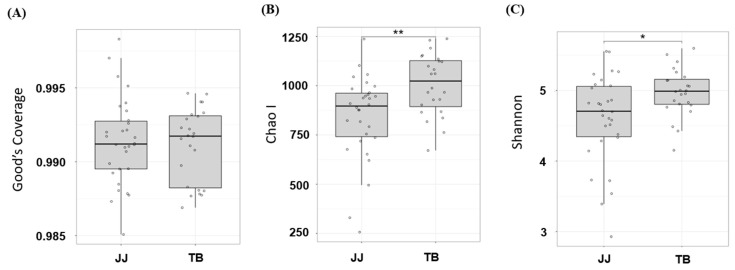
Ecological indices comparison, species richness comparison with Good’s coverage (**A**), Chao Ι (**B**), and species evenness Shannon (**C**); * and ** indicate *p* < 0.05 and *p* < 0.01 with the student’s *T*-test. JJ; Jeju horse, TB; Thoroughbred horse.

**Figure 2 vetsci-08-00081-f002:**
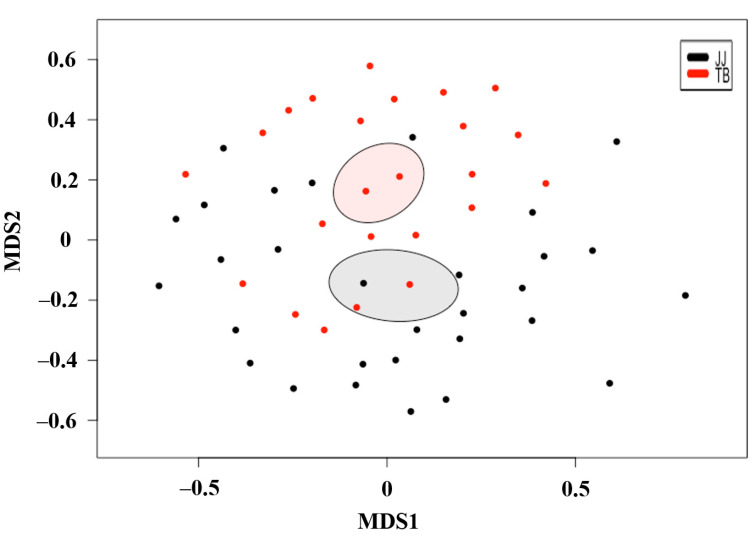
Microbial community comparison based on non-metric multidimensional scaling of Jeju and Thoroughbred horses. Red and black colors indicate microbiota for Jeju and Thoroughbred horses, respectively. JJ; Jeju horse, TB; Thoroughbred horse.

**Figure 3 vetsci-08-00081-f003:**
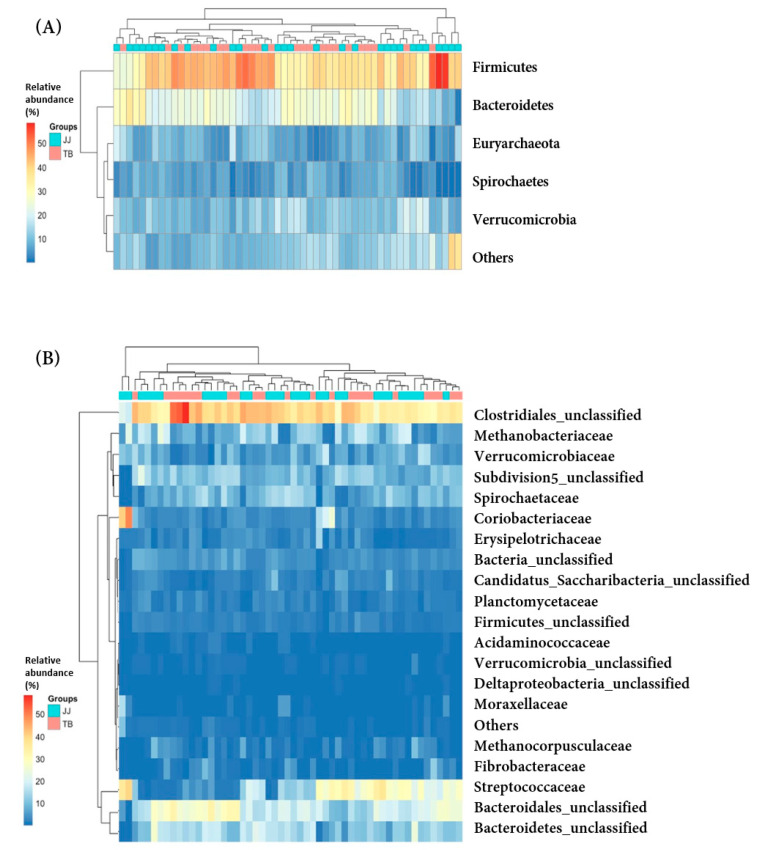
Heatmap profile showing the relative abundance of the phylotype between Jeju and Thoroughbred horses at the phylum (**A**) and family (**B**) levels of Jeju and Thoroughbred horse. JJ; Jeju horse, TB; Thoroughbred horse.

**Figure 4 vetsci-08-00081-f004:**
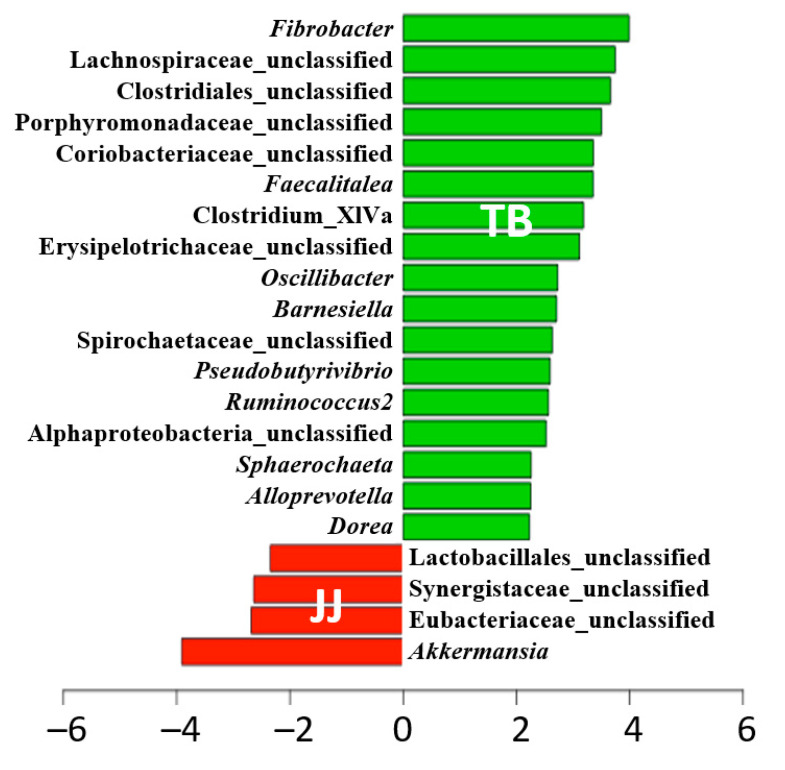
Differentially abundant bacteria based on the LEfSe analysis for Jeju and Thoroughbred horses. JJ; Jeju horse, TB; Thoroughbred horse.

## Data Availability

Publicly available datasets were analyzed in this study. This data can be available in the NCBI SRA database [accession number; PRJNA728810].

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
