# Peer review of "Comparison of the Gut Microbiota of Jeju and Thoroughbred Horses in Korea"

_vetsci, 2021, doi:10.3390/vetsci8050081_

Round 1

Reviewer 1 Report

Overall, this manuscript is very well presented and easy to follow. The knowledge over species microbiota has been a great focus of interest lately. Being horses herbivores that rely on the gut microbiote for their adequate nutrition, it is particular relevant to understand the role of their gut microbiota.    

This said, I would like to comment on a few points I feel that need more clarification:

Lines 25-26:The Thoroughbred horse's gut microbiotas had significantly higher diversity than the Jeju horses”  

I understood this was the case, but I think if it is the main conclusion, it should be more explicit on the abstract. It is not being said that the there is a higher diversity in Thoroughbred horses, but that the two breeds differ on the genera found

Lines 39-40: “A wide variety of microbial ecosystems exist in the gastrointestinal tract of animals, and the gut microbiota plays an essential role in maintaining the health of the host”

Can this be a bit more elaborated? With maybe one or two sentences, establishing the influence of the interaction with microbiota? This comes up on the first paragraph of the discussion, but I think it would help to build a justification in the introduction.

Lines 50-21: “Therefore, the roles of the horse gut microbiota are not completely understood.”

There are some studies describing factors related to gut microbiota in horses. It would be interesting to list some factors here.

Line 92: Here the authors citeChao and Simpson”, but at the results the index presented for evenness is Shannon. Could the authors check if this is correct and include Shannon on the methods?

Discussion: The discussion is well developed, establishing potential factors related to the findings. However, one important point I believe is missing here is the limitations of using this type of samples. In the comparison with other studies, it is also important to establish whether the faecal samples were collected and stored in the same way, as this can interfere in the results. Likewise, even different samples from the same animal could lead to different results. While this is the nature of the method, I think the limitations need to be pointed to balance to which extension the results are comparable.

Lines 231-232 (Conclusion): “These differences in gut microbiota could be owing to types of breeds, feed, and management program [4,20].”

This is not a conclusion from this study. This is a discussion over the results.  

Lines 236-237: “This might be because of the high-quality diet supplied to Thoroughbred horses.”

Would these horses also have a more consistent diet?

Author Response

Response to Reviewer 1 Comments

Point 1: Lines 25-26: “The Thoroughbred horse's gut microbiotas had significantly higher diversity than the Jeju horses”  

I understood this was the case, but I think if it is the main conclusion, it should be more explicit on the abstract. It is not being said that the there is a higher diversity in Thoroughbred horses, but that the two breeds differ on the genera found.

Response 1: Thanks for the comment. According to the alpha-diversity comparison, Thoroughbred horses do have higher diversity microbiota. Therefore, we want to keep the sentence as is. In addition, we would like to take the reviewer’s advice, so we added the following sentence in line 25-27.

  • In addition, beneficial commensal bacteria to produce short-chain fatty acids provide a significant source of energy are also more abundant in Thoroughbred horses.

Point 2: Lines 39-40“A wide variety of microbial ecosystems exist in the gastrointestinal tract of animals, and the gut microbiota plays an essential role in maintaining the health of the host”

Can this be a bit more elaborated? With maybe one or two sentences, establishing the influence of the interaction with microbiota? This comes up on the first paragraph of the discussion, but I think it would help to build a justification in the introduction.

Response 2: We appreciate the suggestions and we added the following modification to the manuscript in line 43-48.

  • The intestinal microbiota performs several essential protective, structural, and metabolic functions for host health, such as digestion of complex host-indigestible polysaccharides, pathogen displacement, and synthesis of vitamins [7,8]. The intestinal microorganisms manufacture short-chain fatty acids (SCFAs) to produce nutrients in the mucus and mucosal membranes and promote the regeneration of intestinal epithelial cells [9,10].

Point 3: Lines 50-51: “Therefore, the roles of the horse gut microbiota are not completely understood.”

There are some studies describing factors related to gut microbiota in horses. It would be interesting to list some factors here.

Response 3: Thanks for the suggestions and we added the following sentence in line 57-59

  • Although several studies had been conducted to investigate the relationship between gut microbiota and digestion [1], disease [2,3,18], and exercise [19,20], the roles of horse gut microbiota are not entirely understood

Point 4:  Line 92: Here the authors cite “Chao and Simpson”, but at the results the index presented for evenness is Shannon. Could the authors check if this is correct and include Shannon on the methods?

 Response 4: Thanks for pointing this out, we made a correction as indicated.

Point 5: Discussion: The discussion is well developed, establishing potential factors related to the findings. However, one important point I believe is missing here is the limitations of using this type of samples. In the comparison with other studies, it is also important to establish whether the faecal samples were collected and stored in the same way, as this can interfere in the results. Likewise, even different samples from the same animal could lead to different results. While this is the nature of the method, I think the limitations need to be pointed to balance to which extension the results are comparable.

 Response 5: Thanks for the suggestions. We added the following paragraph to describe the limitations we faced in this study.  line 249-252

  • In this study, we collected horse feces. Although feces were rectally collected using rectal gloves[22] and stored immediately on ice, storage condition may have affected the microbial community, as previously reported [50]. Moreover, multiple sampling for the same horses could minimize inter-individual deviation.

Point 6: Lines 231-232 (Conclusion): “These differences in gut microbiota could be owing to types of breeds, feed, and management program [4,20].”

This is not a conclusion from this study. This is a discussion over the results.  

 Response 6: We appreciate your comment and we moved the sentence to the discussion (line 241-242).

Point 7: Lines 236-237: “This might be because of the high-quality diet supplied to Thoroughbred horses.”

Would these horses also have a more consistent diet?

Response 7:  Yes, they were provided with more consistent diet. To make this clear, we added the following sentence: line 265-266

  • The diet contained well-adjusted, and balanced nutrients with essential vitamins, which were consistently provided.

Reviewer 2 Report

The manuscript titled ‘ Comparison of the gut microbiota of Jeju and Thoroughbred Horses in Korea’ by Park et al. describes the gut microbiome of two horse breeds in Korea. Overall, the manuscript is well written however, needs some more work. My comments are as follows.

  1. The last two paragraphs in the introduction need references. The authors should also comment on if these horse breeds are economically important in the introduction. There is not enough background provided that justifies studying the microbiome of these horses.
  2. Were rectal swabs taken in addition to fecal samples ?
  3. The conclusion from figure two is unclear. What are the two circles representing ? Is there any grouping ?
  4. The figure legends in figure three are not clear. What are the colors on the heat map represent ?
  5. The main issue with study is the authors do not consider multiple variables that can affect the gut microbiome. For each of the horse analyzed more variables e.g the diet, age, location, breed, gender must be included. The authors only analyzed the impact of breed on the microbiome without considering other variables. This needs to be reanalyzed and a non-metric multidimensional scaling (NMDS) must be plotted to check for any clustering.

Author Response

Response to Reviewer 2 Comments

Point 1: The last two paragraphs in the introduction need references.

Response 1Thanks for your comments and we added references to the two paragraphs as suggested (line 51-68)

  • The study of gut microbiota has generally been conducted using culture-dependent methods [14,15].
  • These methods are limited in investigating the horse gut microbiota because approximately 70% of it cannot be cultured in the laboratory [16].
  • The development of next-generation sequencing (NGS) and bioinformatic tools has overcome the limitations of traditional culture-dependent methods [4,17].
  • Although several studies had been conducted to investigate the relationship between the gut microbiota and digestion [1], disease [2,3,18], and exercise [19,20],
  • However, as widely acknowledged, the microbiota plays a vital role in maintaining the health of the host, including horses [18].
  • Thoroughbred horses are sprint racehorses that have adjusted to the environment in Jeju Island over a short period [21].

Point 2: The authors should also comment on if these horse breeds are economically important in the introduction. There is not enough background provided that justifies studying the microbiome of these horses.

Response 2We appreciate the comments and added the following sentences (line 61-64)

  • The Jeju horse is the only indigenous Korean horse breed since the Stone Age. They are inherently rare. Thus, they are conserved well and designated as Korean Natural Monument 347. Recently, Jeju horses have been used as racing horses; thus, the economic value of Jeju horses has been increased close to that of Thoroughbred horses.

Point 3: Were rectal swabs taken in addition to fecal samples ?

Response 3Fecal samples were obtained directly from rectum using rectal gloves. We modified the manuscript to mention this in line 90-92.

  • Fecal sampling was performed from the rectum using rectal gloves with sterile lubrication (Kruuse, Denmark) to reduce environmental contamination, as described previously [22].

Point 4: The conclusion from figure two is unclear. What are the two circles representing ? Is there any grouping ?

Response 4: We appreciate the comments. The colors represent JJ and TB microbiota which is indicated at the top right corner within the figure. The ellipses were drawn based on 95% confidence calculated by vegan R package, which is mentioned in the method section regarding to the non-metric multidimensional scaling analysis. We added color indication to the figure legend as follows: “Red and black colors indicate each microbiota for Jeju and Thoroughbred horses, respectively. “

Point 5: The figure legends in figure three are not clear. What are the colors on the heat map represent ?

Response 5: We appreciate the comments. The colors are for the relative abundance of each taxon, where red and blue indicate high and low abundance, respectively. To clearly indicate, we added “relative abundance (%)” above the color key in the figure.

Point 6: The main issue with the study is the authors do not consider multiple variables that can affect the gut microbiome. For each of the horse analyzed, more variables e.g the diet, age, location, breed, gender must be included. The authors only analyzed the impact of breed on the microbiome without considering other variables. This needs to be reanalyzed and a non-metric multidimensional scaling (NMDS) must be plotted to check for any clustering.

Response 6:  We appreciate the comments. As the reviewer suggested that there are several factors that could influence the gut microbiota other than breed. However, we selected horses that are similar age and weight, body condition scoring (BCS), soundness, vaccination, deworm and medication.  We added this  to the method section regarding to the selection of horse subjects. line 82-85

  • All horses had not experienced any changes in their diet and housing conditions in the recent three months and were carefully selected for minimizing the variations in age, weight, body condition scoring (BCS), soundness, vaccination, deworm, and medication.

We also added the following paragraph to describe the limitations we faced in this study in line 249-252

  • In this study, we collected horse feces. Although feces were rectally collected using rectal gloves [22] and stored immediately on ice, storage conditions may have affected the microbial community, as previously reported [12]. Moreover, multiple sampling for the same horses could minimize inter-individual deviation.

Reviewer 3 Report

The study is well designed and the manuscript well written, few very minor comments:

Line 45: These methods are…

Line 46: 70% of it cannot…

Line 284: 2018 (as it should be bold)

Author Response

Response to Reviewer 3 Comments

Point:  The study is well designed and the manuscript well written, few very minor comments:

Line 45: These methods are…

Line 46: 70% of it cannot…

Line 284: 2018 (as it should be bold)

Response:

We appreciate the corrections. As reviewer suggested, we have modified the manuscript as follows:

Line 52-53

  • These methods are limited in investigating the horse gut microbiota because approximately 70% of it cannot be cultured in the laboratory [16].

Line 336-337

  • Douglas, G.M.; Beiko, R.G.; Langille, M.G. Predicting the functional potential of the microbiome from marker genes using PICRUSt. In Microbiome Analysis; Springer: New York, USA, 2018; pp. 169-177.

Reviewer 4 Report

The article is interesting because the authors include data from an unusual breed. However, the results can not be explain just for the differences between breeds. There are many other factors that can provide different probiota in horses. The article can be improved if data from vaccination and deworming status are included in the material and methods section. Also it would be interesting to include in the discussion other theories to explain the results, not just the breed (e.g. type of exercise, vaccines, deworming programs, specific type of hay suministered...). 

Author Response

Response to Reviewer 4 Comments

Point 1: The article is interesting because the authors include data from an unusual breed. However, the results can not be explained just for the differences between breeds. There are many other factors that can provide different microbiota in horses. The article can be improved if data from vaccination and deworming status are included in the material and methods section.

Response 1:  We appreciate the comments. As the reviewer suggested that there are several factors that could influence the gut microbiota other than breed. However, we selected horses that are similar age and weight, body condition scoring (BCS), soundness, vaccination, deworm and medication.  We added this to the method section regarding to the selection of horse subjects. line 82-85

  • All horses had not experienced any changes in their diet and housing conditions in the recent three months and were carefully selected for minimizing the variations in age, weight, body condition scoring (BCS), soundness, vaccination, deworm, and medication.

Point 2: Also, it would be interesting to include in the discussion other theories to explain the results, not just the breed (e.g. type of exercise, vaccines, deworming programs, specific type of hay suministered...). 

Response 2:  We appreciate the comments. In discussion, we mentioned other factors that could influence the horse microbiota including those mentioned here. Please see line 242-248.

  • Previous studies have reported that parasite, exercise, age, and stress significantly shifts horse gut microbiota [19,47-49]. Exercise and aging increased the abundance of Firmicutes while stress decreases the abundance of Firmicutes [19,20,49]. Various factors could affect the abundance of several genera. Although we minimized these factors by selecting horses similar in physiology, it is recommended that future studies include a bigger sample size and various physiological and environmental parameters.

Round 2

Reviewer 2 Report

The authors have responded to all the queries.